



# Time-dependent modeling of Alfvénic precipitation observed in the ionosphere

Etienne Gavazzi[1], Andres Spicher[1], Björn Gustavsson[1], James Clemmons[2], Robert Pfaff[3], and Douglas Rowland[3]

[1]Department of Physics and Technology, UiT The Arctic University of Norway, Tromsø, Norway
[2]Department of Physics and Astronomy, University of New Hampshire, Durham, NH, USA
[3]NASA Goddard Space Flight Center, Greenbelt, MD, United States

**Correspondence:** Etienne Gavazzi (etienne.gavazzi@uit.no)

**Abstract.** Small scale dynamic auroras are generally related with dispersive Alfvén waves. Due to the short spatial and time scales involved, investigating the effects of this type of auroral precipitation on the ionosphere is challenging. In this study, we address this challenge by introducing a recently developed and improved time-dependent electron transport code entitled AURORA. We use high-resolution data from the Visualizing Ion Outflow via Neutral Atom Sensing-2 (VISIONS-2) sound-

ing rocket campaign as input for the modeling. The rocket flew through the active dayside auroral region and the onboard instrumentation measured signatures of Alfvénic precipitation varying on sub-second timescales. With the code, we model the propagation of the electron flux in the ionosphere and we provide a first-order validation for the case studied here. We then present two examples illustrating the modeling capabilities by showing ionization and optical emission rates for Alfvénic and mono-energetic precipitation with similar downward energy flux. The model results show variations in the height of maximum

ionization from about 120 km to 180 km in less than 0.3 s for the Alfvén case, while it remains stable at about 160 km for the mono-energetic case. Additionally, For Alfvénic precipitation, the modeled intensities exhibit a short lived peak at 6730 Å and 4278 Å, while for the latter case, the intensities are constant and dominated by 6730 Å and 8446 Å emissions. The modeling introduced here opens for possibilities to further advance our understanding of small scale dynamic aurora.

## 1 Introduction

The high latitude ionosphere is a complex and variable region influenced by its coupling to the magnetosphere and the thermosphere (e.g. Kelley, 2009). Visible consequences of the coupling with the magnetosphere are the auroras. Auroras are primarily caused by the precipitation of electrons with typical energies ranging from hundreds of eV to several keV that have been accelerated along magnetic field lines from the magnetosphere down into the ionosphere (Knudsen et al., 2021). The different auroral forms cover a wide range of spatial scales from hundred of meters to several tens kilometers in latitude and thousands of

kilometers in longitude, and can be relatively stable or very dynamic in time. These spatio-temporal characteristics are thought to mirror the phenomena and conditions in the magnetosphere that are driving the electron precipitation (e.g. Lysak et al., 2020; Kataoka et al., 2021). Accordingly, auroras are often classified as quiet discrete arcs (e.g. Karlsson et al., 2020; Lysak et al.,





2020, and references therein) or as small-scale dynamic auroras (e.g. Kataoka et al., 2021, and references therein), and some of their main characteristics are repeated below.

Large scale stable auroras are often referred to as *quiet discrete arcs*. These are typically east–west elongated, with a north–south width on the order of 1 to 10's of km and can be stable for tens of minutes (Karlsson et al., 2020). They are associated with quasi-static electric potential structures along the magnetic field lines (Birn et al., 2012; Karlsson et al., 2020; Lysak et al., 2020). The resulting precipitation typically consists of mono-energetic high-energy electrons with a low-energy tail of secondary electrons that have been mirrored back down by the electrical fields. Time–energy spectrograms of the electron flux measured by spacecrafts moving rapidly through the region located under the electric potential show structures with an "inverted-V" shape (Frank and Ackerson, 1971; Gurnett and Frank, 1973; Lysak et al., 2020).

    At the other end of the temporal and spatial scale are *small-scale dynamic auroras*. These typically span hundreds of meters to a few kilometers, and evolve within a few seconds (Kataoka et al., 2021). Examples are flickering auroras (e.g. Gustavsson et al., 2008; Whiter et al., 2010), vortices (e.g. Trondsen and Cogger, 1998), filaments (e.g. Dahlgren et al., 2008, 2013), etc.
(Sandahl et al., 2011; Kataoka et al., 2021). These types of auroras are thought to be related to interactions of dispersive Alfvén waves (DAWs) responsible for the acceleration of the electrons (Semeter et al., 2008; Kataoka et al., 2021). Observations realized with the FAST satellite indicate that DAWs seem to be the dominant acceleration mechanism on the poleward edge of the aurora around noon and pre-midnight (Chaston et al., 2007). In time–energy spectrograms of the electron flux, the associated precipitation is seen to be broadband in energy (Stasiewicz et al., 2000; Colpitts et al., 2013; Kataoka et al., 2021).
Measurements with high time-resolution show that the broadband energy structures are often dispersed in time (Arnoldy et al., 1999; Andersson et al., 2002; Tanaka et al., 2005), a feature reproduced by modeling of acceleration by Alfvén waves (Kletzing and Hu, 2001). DAWs and the associated auroral precipitation vary on such short spatial and temporal scales that observations and modeling are challenging. Therefore, they and their auroral impact are less well understood compared to discrete auroral arcs (Sandahl et al., 2011; McCrea et al., 2015). Consequently, several questions remain open about their
effects on the ionosphere.

    Auroral precipitation is an important source of ionization at high-latitudes. It can change ionospheric parameters such as the plasma density (Labelle et al., 1989; Moen et al., 2002, 2013; Kaeppler et al., 2015; Buschmann et al., 2023), the electron temperature (Lynch et al., 2007) or the conductivity (Reiff, 1984; Lysak, 1990; Cowley, 2000; Kaeppler et al., 2015; Yu et al., 2022). This can in turn affect the coupling between the magnetosphere–ionosphere–thermosphere (MIT) systems and impact
processes such as plasma convection (Labelle et al., 1989; Moen et al., 2013), ion outflow and heating (Lynch et al., 2007) or the ionospheric feedback instability (IFI) mechanism (Lysak and Dum, 1983; Streltsov and Lotko, 2008; Cohen et al., 2013).

    To study the impact of auroral precipitation on the ionosphere, several methods can be used. Historically, range–energy deposition methods were used (e.g. Rees, 1963). With time, electron transport methods were developed (e.g. Strickland et al., 1976; Stamnes, 1981; Lummerzheim and Lilensten, 1994; Solomon, 2017). This type of code takes an incoming precipitating
electron flux at the top of the ionosphere and models the propagation of the supra-thermal electron in the ionosphere and their interaction with the neutral atoms and molecules. The primary output of these transport models are the supra-thermal electron flux along a magnetic field line. From such flux, it is possible to calculate profiles of ionization rates (e.g. Lummerzheim





and Lilensten, 1994; Kaeppler et al., 2015), heating rates (Lynch et al., 2007), and auroral volume emission rates at different wavelengths (e.g. Strickland et al., 1989; Hecht et al., 1989; Meier et al., 1989; Lummerzheim and Lilensten, 1994). The

latter is often used in combination with optical measurements to determine properties about the ionosphere and/or about the precipitation (Hecht et al., 1989; Meier et al., 1989; Lanchester et al., 2009; Whiter et al., 2010; Grubbs II et al., 2018; Gabrielse et al., 2021). The models cited above use a steady-state approximation, which is sufficient to study relatively stable auroral precipitation, such as quiet discrete arcs.

However, the steady-state approximation is no longer valid when the precipitation varies on short enough timescales, when

the different electron time-of-flight effects becomes important, which is estimated to be around half a second (Sandahl et al., 2011). This is longer than the typical time-scales of variations seen in small-scale dynamic aurora (Kataoka et al., 2021). Thus, Sandahl et al. (2011) identified the need for time-dependent electron transport models in order to advance our understanding on small-scale dynamic precipitation and its effects on the ionosphere and the whole MIT coupling. Time-dependent multi-stream models are much more computationally heavy than steady-states models, and the first time-dependent electron transport code

for auroral precipitation (Peticolas and Lummerzheim, 2000) had to simplify the problem to a one-stream model with primary electron flux in the field-aligned direction, with not scattering, and with not propagation of the secondary electrons. As noted by Sandahl et al. (2011), this makes it challenging to study the precise spatio-temporal evolution of auroral electron flux in the ionosphere.

In this study, we combine sounding rocket observations with the Julia implementation of AURORA (Gavazzi and Gustavs-

son, 2025), a recently developed time-dependent electron transport code (Gustavsson, 2022) to model auroral precipitation on sub-second time-scales. We use high-resolution data from the Visualizing Ion Outflow via Neutral Atom Sensing-2 (VISIONS-2) sounding rocket campaign that flew through the dynamic dayside aurora (Takahashi et al., 2022) and identified small-scale dynamic auroral precipitation. The measured downward electron flux vary on 50 ms time-scales, and are input to AURORA which models the time-varying electron flux in the ionosphere. This makes it possible to model the impact of small-scale au-

roras. Here, we present the calculated ionization and optical emission rates associated with dispersed electron signatures and compare them to stable mono-energetic precipitation.

The outline of the paper is the following. In section 2 we present the rocket instrumentation and the AURORA model. In section 3 we describe the events observed in the rocket data and used in this study. We present the electron flux simulated by the code and show different profiles in time of the ionization and optical emission rates from the two classes of precipitation

mentioned above. We then proceed with a first-order validation of AURORA through a comparison with in-situ measurements. Finally, we summarize our results and discuss future work in section 4.

## 2   Instrumentation and simulation

In this study, we use data from the VISIONS-2 sounding rocket mission. VISIONS-2 consists of two sounding rockets that were launched from Ny-Ålesund, Svalbard, on 7 December 2018 at 11:06:00 and 11:08:00 UTC. The two rockets flew through

the active dayside auroral region, and reached apogees of 806.6 and 601.2 km (Takahashi et al., 2022).





We use data from the electric field and magnetic field instruments onboard the low-flyer, and focus on the electron flux data obtained from a top-hat electrostatic analyzer (ESA) also mounted on the low-flyer. The top-hat ESA measures the electron energy flux in time, energy and pitch-angle. The time resolution of the measurements is of 50 ms, and the electrons are measured with energies from 3 eV to 30 keV. The instrument has 20 different channels that measure the electron energy flux with different pitch-angles for each channel. The pitch-angles being measured span from 0 to 360°, where 0° is field-aligned down and 180° is field-aligned up. Due to the circular symmetry of the gyration of the electrons along magnetic field lines, we only need one side of the measurements. As such, we average the electron flux measurements from 0–180° with the measurements from 180–360° when using the rocket electron flux data in this study. This has the benefit to mitigate instrumental challenges, such as dead channels (no measurements) or the shadowing of some channels by the rocket structure leading to systematically lower measurement intensity.

The rocket provides point measurements of the electron flux along its trajectory. To investigate the effects of the precipitation structures observed in the rocket data on the ionosphere, we use an electron transport code to calculate the propagation of the measured electron flux under the rocket along a magnetic field line. Since the instrument detects rapid variations of the electron precipitation on time scale faster than the electron time-of-flight through the ionosphere, it is necessary to use a time-dependent electron transport code. We use AURORA, a time-dependent multi-stream electron transport code used and published for the first time in Gustavsson (2022). Recently, the code has been improved and re-implemented in the Julia programming language (Bezanson et al., 2017), leading to both reduced run-time of the simulations and more accurate electron flux results. The code is open-source, and the version of the code used to produce the results shown in this article is available at Gavazzi and Gustavsson (2025).

A proper description of how the code works is given in Gustavsson (2022) and Gavazzi (2022). Key points are that the code is multi-stream and time-dependent. Multi-stream capabilities are necessary for time-dependent calculations, as electrons with different pitch-angles will have different field-aligned velocity components. The code takes electron number flux as a function of energy, pitch-angle and time at the top of the ionosphere as input, and propagates the electrons along a magnetic field line (1D), taking into account time-of-flight effects. This is done by solving the system of electron transport equations

$$\frac{1}{v(E)}\frac{\partial I_e(z,\theta,E,t)}{\partial t} + \cos\theta\frac{\partial I_e(z,\theta,E,t)}{\partial z} = \left(\frac{\partial I_e(z,\theta,E,t)}{\partial t}\right)_{coll}, \tag{1}$$

with a Crank–Nicholson scheme.

Acceleration from electric fields and magnetic mirroring effects are ignored. This is common to most electron transport codes and is based on the assumption that collisions are dominating the physics (Strickland et al., 1976; Solomon, 1987, 2017; Lummerzheim and Lilensten, 1994; Peticolas and Lummerzheim, 2000). The effects of collisions are present on the right-hand side of the transport equations. AURORA takes into account elastic, inelastic and ionization collisions, and we run it with the three major neutral species $N_2$, $O_2$ and $O$. The neutral densities are extracted from the NRLMSIS 2.1 model (Emmert et al., 2021). Collisions with thermal electrons are also modeled. The thermal electron densities and temperatures are taken from the IRI2016 model (Bilitza et al., 2017). The code degrades the primary electrons in energy and produces secondary electrons isotropically.



AURORA is run with the observed electron flux, $I_e(E, t, \theta)$ as input at the highest altitude. 18 pitch-angle streams uniformly spaced from 0° to 180°, each with a width of 10°, were used to best model the pitch-angle distribution. The energy and pitch-angle grid of the observations were up-sampled to match the default grid of AURORA. The $I_e$ observations were made with a time-resolution of 50 ms and are taken as constant over each interval. To properly resolve time-of-flight effects for the fastest electrons, at 5 keV, AURORA integrates the transport equations with time-steps smaller than 500 μs.

From the ionospheric flux $I_e(z, \theta, E, t)$ produced by AURORA, one can calculate the evolution in time and height of quantities that depend on the electron flux. For example, profiles of the ionization rate in time are given by

$$q_{ionization}(z, t) = \sum_k n_k(z) \sum_j \int_{E_j}^{E_{max}} I_e(z, E, t) \sigma_k^j(E) dE, \tag{2}$$

where $\sigma_j^k(E)$ is the cross section for the ionization reaction $j$ of the $k$-th species for collision of electron with energy $E$. The ionization reactions include ionization to excited states, dissociative ionization and double ionization. $I_e(z, E, t)$ is the electron

number flux integrated over the different pitch-angle streams. $E_j$ is the excitation threshold of the ionization reaction $j$.

Similarly, one can calculate the excitation rates of different excited states of the neutrals, given the corresponding excitation cross-sections. If the relaxation from these states lead to prompt optical emissions, then the excitation rates are equivalent to optical emission rates. If the excited states have non-negligible lifetimes, the full ion chemistry with effects from radiative decay, quenching, diffusion, drift, etc. should also be taken into account to obtain the optical emission rates. This is out of the

scope of this article, and here we limit ourselves to prompt emissions.

## 3    Analysis

In this section, the rocket data and the selected events are presented. Further, we show the ionospheric electron flux modeled with AURORA with the in-situ observations of downward electron flux as input. We then apply the model to two events, one with a short burst of time–energy dispersed precipitation and one with steady mono-energetic precipitation, and compare the

volume ionization and emission rates for the two cases. Finally, we perform a first-order validation of the code by comparing the simulated upward electron flux with the in-situ observations.

### 3.1    In-situ observations

The rocket data for the two time intervals of interest are shown in Fig. 1. The perpendicular electric and magnetic fields are shown in row (a). The DC components have been filtered out to leave only the fluctuations. The field-aligned Poynting flux is

shown in row (b). Positive values correspond to a downward flux into the ionosphere. The perpendicular electric field vectors are shown in row (c). These are down-sampled and plotted every 20 ms to make the plot less cluttered. Positive y-direction corresponds to the geographic north, and positive x-axis corresponds to the geographic east. The cross-power spectra between the perpendicular electric and magnetic fields is shown in row (d). The field-aligned (pitch-angle of 0–10°) precipitating



**Figure 1.** Rocket measurements for two time intervals. (a) Northward and eastward components components of the electric and magnetic fields. The DC components have been filtered out. (b) Field-aligned Poynting flux, with positive values corresponding to a downward flux into the ionosphere. (c) Perpendicular electric field vectors. Each arrow is one measurement in time, but not all measurements are shown to reduce cluttering. Positive y-direction corresponds to the north, and positive x-direction corresponds to the east. (d) Cross-power between the perpendicular electric and magnetic fields. (e) Precipitating electron energy flux with a pitch-angle between 0 and 10° (approximately field-aligned). (f) Field-aligned incoming (yellow) and outgoing (blue) total electron energy flux (dashed lines) and total electron number flux (full lines), as well as the altitude of the rocket. The grey boxes with magenta borders and with the names `aw1`, `aw2` and `inv` highlight the precipitation events used in this study.





electron energy flux measured by the top-hat ESA is presented in row (e). The downward and upward field-aligned electron
number flux and electron energy flux are presented in row (f). The altitude of the rocket is also plotted in row (f).

Thanks to the high temporal resolution of the electron energy flux measurements, we are able to distinguish several time–energy dispersed structures in row (e) of Fig. 1, most notably around 475 s. They are short in time, with a duration of around 0.5 s per structure. The flux extend over a wide range of energies, with high energies arriving first and the lower energies progressively coming later. This type of time–energy dispersed precipitation is thought to be caused by the acceleration of
electrons by dispersive Alfvén waves (Kletzing and Hu, 2001; Andersson et al., 2002). Many of these structures are present in the rest of the whole flight data. They are often seen to come as a group, like a train of wave-crests.

These structures are seen clearly in row (e) of Fig. 1 between 474 s and 476 s, as well as between 535.5 s and 537 s. They correlate with intense perturbations in the perpendicular electric and magnetic fields (row (a) of Fig. 1), reaching 100 mV/m for the electric field and 50 nT for the magnetic field. The simultaneous intense perturbations in the electric and magnetic fields
are clearly visible in the cross-power spectra (row (d) of Fig. 1)), with high cross-power from 1 to almost 15 Hz. These intense fluctuations in the perpendicular field components can be a sign of the presence of dispersive Alfvén waves (DAW) (Stasiewicz et al., 2000; Miles et al., 2018; Pakhotin et al., 2020). The strong correlation of the electric and magnetic field fluctuations also shows as a strong downward Poynting flux as seen in row (b) of Fig 1. This indicates an important electromagnetic energy-flux into the ionosphere, associated with the DAWs and their dispersed precipitation. At the same time, vortex-like structures are
seen in the perpendicular electric field shown in the row (c) of Fig. 1. These vortices are believed to be caused by DAWs (Stasiewicz et al., 2000). The high cross-power, high downward Poynting flux and the vortices associated with the dispersed structures in electron flux are to compare to the nearly zero cross-power, zero Poynting flux and the absence of vortices from 478 s to 482 s, a time interval in which the electron precipitation is mono-energetic. The difference between the dispersed and mono-energetic nature of the electron precipitation is also clearly visible in the total parallel electron number- and energy- flux
plotted in row (f) of Fig. 1, where the dispersed structures come with net peaks in both the up- and down-ward parallel flux, whereas the mono-energetic precipitation displays a very constant total parallel flux.

The electron flux used in this study are taken from the time intervals of data shown in Fig. 1. In particular, we run the model with the three precipitation events highlighted with the magenta boxes. The first event comprises the dispersed structure seen at around 475 s. This interval will be referred to as the `aw1` event. The second event consists of constant precipitation seen
between 480 s and 481 s, and will be referred to as `inv`. These two events (`aw1` and `inv`) are selected for a comparison between dispersed electron signatures and mono-energetic precipitation in subsection 3.3. These two events are representative for the two types of precipitation and are measured by the rocket close in time and at nearly the same altitude. The third interval includes the dispersed structure seen around 536 s, hereafter entitled `aw2`. This event is used to perform a validation of AURORA in subsection 3.4. We choose this event as it is relatively isolated in time from other precipitation, which is ideal
for performing the comparison with in-situ data.




**Figure 2.** The top panel presents the downward electron energy flux of the `aw1` event used as input. Vertical dashed lines correspond to the times selected for the rest of the figure. Rows (a), (b) and (c) show snapshots of the electron energy flux $I_{eE}$ calculated by the transport code at 0.041 s, 0.129 s and 0.323 s after the start of the precipitation/simulation, respectively. The different columns correspond to different pitch-angle streams.




## 3.2 Ionospheric electron flux

The flux of energetic electrons in the ionosphere resulting from the downward flux observed with the top-hat ESA have been calculated with AURORA. Here, we present the flux resulting from the `aw1` event. In Fig. 2 are shown snapshots at 0.041 s, 0.129 s and 0.323 s after the start of the event. Here we have merged pitch-angle streams (e.g. 10–20° and 20–30° have been merged), and only the downward flux are shown. An animation of the simulation results showing both down- and up-ward electron flux is available in the supplementary material.

In the first snapshot at 0.041 s after the start of the precipitation (row (a) of Fig. 2), just before the start of the time–energy dispersed structure, it is obvious that electrons with a small pitch-angle reach low altitudes much faster than electrons with a larger pitch-angle. Within a given pitch-angle stream, we can also observe that the electrons with a high energy are propagating faster than the electrons with a lower energy, as is expected from $v = \sqrt{2E/m}$. In that first snapshot, the high energy field-aligned electrons have already reached low altitudes in the ionosphere ($< 200$ km) where neutral densities are higher and started to produce secondary electrons in ionization collisions. This secondary electron production is visible as a nearly isotropic flux of electrons at low energies ($< 20$ eV).

The production of secondary electrons continues in the second snapshot at 0.129 s (row (b) of Fig. 2), as an increasing number of primary electrons are reaching the lower ionosphere. In the animation available as supplementary material, one can observe the secondaries streaming up along the field line.

At the time of the third snapshot, at 0.323 s (row (c) of Fig. 2), the low-energy end of the time–energy dispersed structure `aw1` is arriving at the top of the ionosphere. In the large pitch-angle streams, such as panels 3, 4 and 5 of the row (c), the dispersion in time and energy of the precipitation is visible in the ionospheric flux. Indeed, the high energy electrons ($> 100$ eV) are seen to reach altitudes below 300 km, while the lower energy part of the precipitation ($< 100$ eV) is just arriving at the top of the modeled ionosphere, above 400 km. This is less clear for more field-aligned pitch-angle streams such as in panel 1 and 2 of row (c), as the electrons with a lower pitch-angle have shorter path-lengths and thus shorter time-of-flights through the ionosphere. In consequence, these streams follow the variations in the observed precipitation than the streams with higher pitch-angles. Also visible in this third snapshot at 0.323 s is the further increased intensity of the flux of secondary electrons in all pitch-angle streams. In the animation available as supplementary material, it is possible to see in the upward flux that some of the secondaries that were produced at an earlier time as well as some of the scattered primary electrons at higher energy are reaching the top-boundary of the simulation, which corresponds to the rocket altitude at the time of measurement of the `aw1` event used here as precipitation.

## 3.3 Comparison of Alfvénic and mono-energetic precipitation

In this section we compare the ionospheric response to one time–energy dispersed structure and a case with persistent mono-energetic precipitation. Volume ionization rates, volume emission rates and column-excitation rates are compared.

We run and compare the `aw1` and `inv` events. The `inv` event was measured by the rocket five seconds after the `aw1` event. The `aw1` was measured by the rocket at an altitude of 595 km while the `inv` event was measured at an altitude of 593 km.





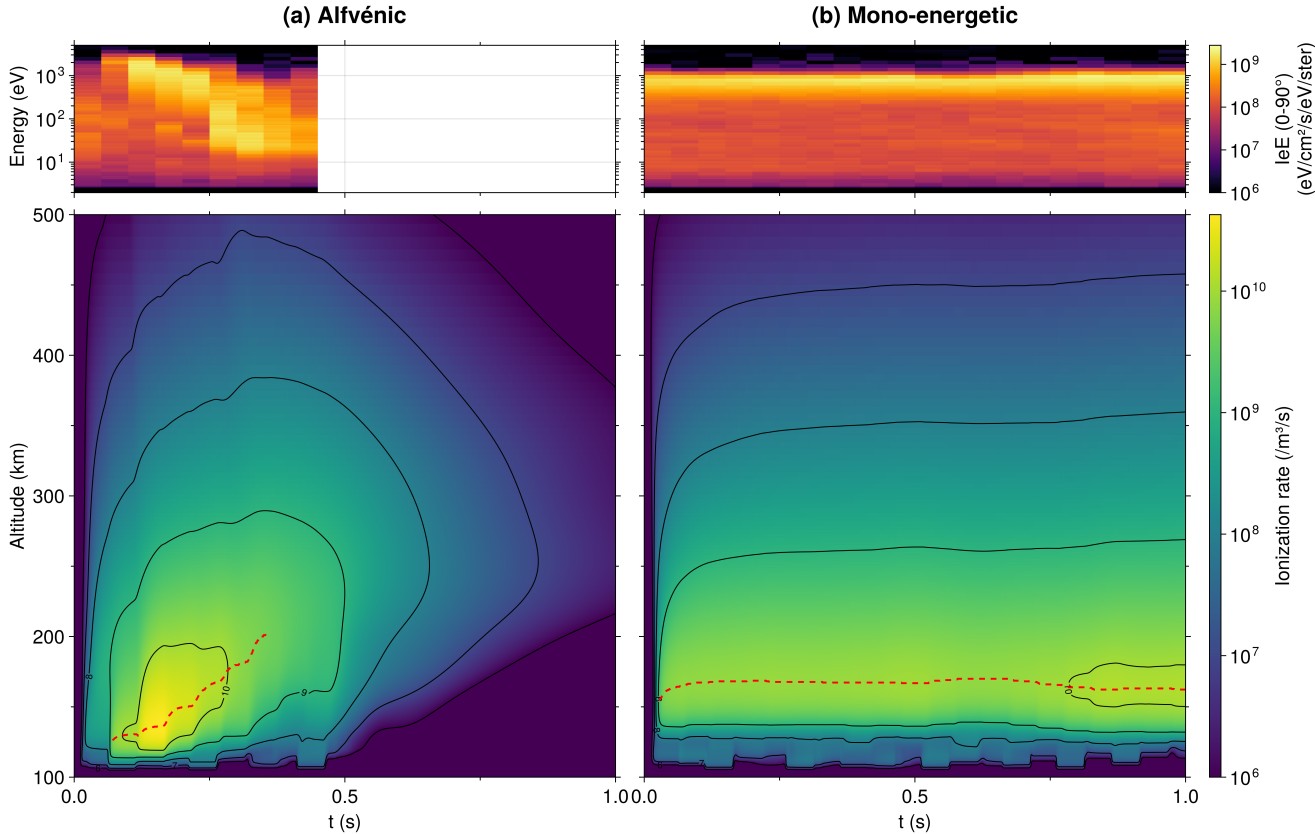

**Figure 3.** Ionization rates in time and height as calculated by the transport code. The top panels show the precipitation used as input. The bottom panels show the calculated ionization rate profiles. (a) Using the dispersed structure in precipitation from the `aw1` event as input. (b) Using the mono-energetic precipitation from the `inv` event as input. For both event, the red dashed line indicate the height of maximum ionization rate, as long as the value is greater than 10% of the maximum ionization rate value for the whole event.

As can be seen in Fig. 1f, the two intervals exhibit a similar incoming energy flux, but a very different shape of precipitation. The `aw1` event presents a time–energy dispersed structure in precipitation, part of what looks like a succession of Alfvénic structures. The `inv` event consists of a subset of a mono-energetic structure stable for several seconds.

As described in the section 2, we can calculate profiles of the ionization rate from the modeled electron flux $I_e$ using Eq. (2). This is done for both the `aw1` and `inv` events, and the resulting ionization rates are shown in Fig. 3. The peak intensity of the ionization rate in response to the `aw1` precipitation (Fig. 3a) moves up in altitude with time, from around 125 km to 200 km in slightly less than 0.3 s. This is to be compared to the ionization rate produced by the `inv` precipitation (Fig. 3b), where the highest intensity remains at the same altitude, at around 160 km. The simulation for the `aw1` event was run for an extra 0.55 s without any precipitation to show what is happening after the precipitation stops. We observe that at low (< 200 km) and high





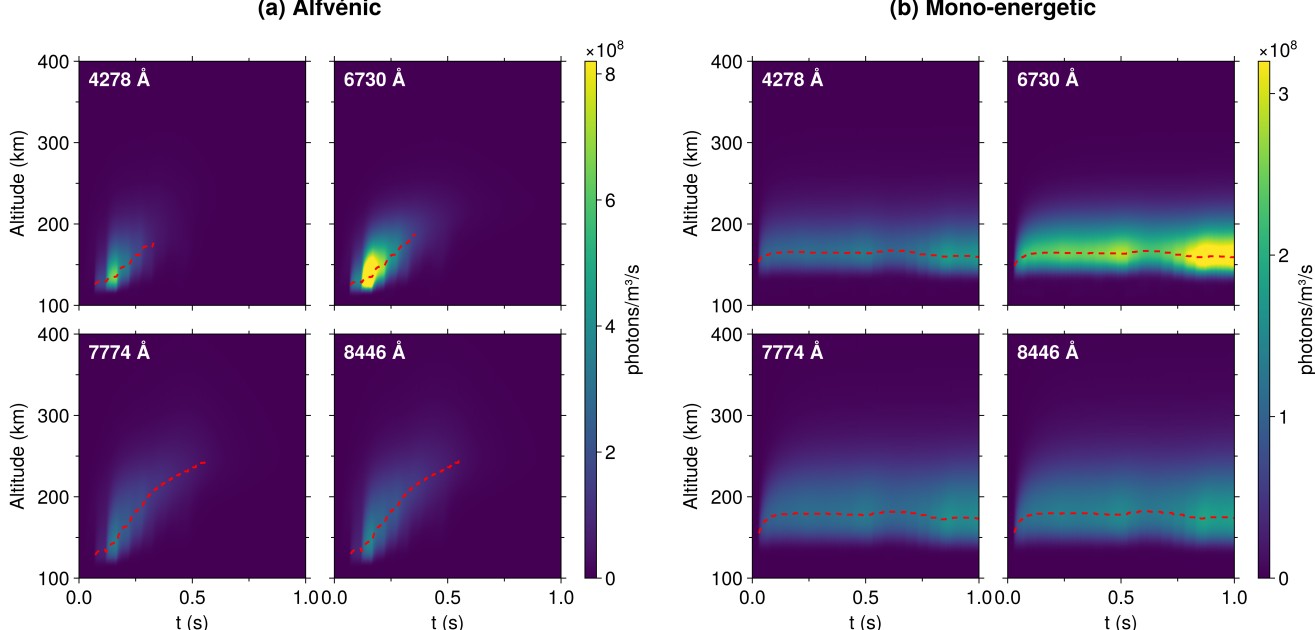

**Figure 4.** Volume emission rates in time and height as calculated by the transport code. For each event are plotted in sub-panels the volume emission rates at 4278 Å emitted by $N_2^+(B^2\Sigma_u^+)$, 6730 Å by $N_2(B^3)$, 7774 Å by O and 8446 Å by O. (a) Emission rates calculated using the dispersed precipitation from the `aw1` event as input. (b) Emission rates calculated using the mono-energetic precipitation from the `inv` event as input. The red dashed line in each panel indicates the height of maximum emission, as long as the value is greater than 10% of the maximum emission value for the whole event.

($> 400$ km) altitudes, the ionization rate quickly decreases to values under $10^6 \ ionization/m^3/s$, while it decreases more slowly at altitudes between 200 and 400 km.

The optical volume emission and excitation rates are obtained with Eq. (2) using the corresponding emission and excitation cross-sections. These are shown in Fig. 4. Similarly to the ionization rates shown in Fig. 3, the optical emission rates are seen to vary systematically in height for the Alfvénic precipitation from event `aw1`, while they are stable in height for the mono-energetic precipitation from event `inv`. For the `aw1` event, the peak-altitude of the 4278 Å and 6730 Å emissions, from $N_2^+$ and $N_2$ respectively, shift from around 120 km to 180 km in less than 0.3 s, while for the `inv` event, the peak-altitude of the

same emissions is stable at around 160 km of altitude. For the `aw1` event again, the peak-altitude of the 7774 Å and 8446 Å emissions, from O, vary over a larger altitude range and persist for a longer time after the end of the primary precipitation than the 4278 Å and 6730 Å emissions. They are seen to vary from around 120 km to 250 km in a bit less than 0.5 s. For the `inv` event, the peak-altitude of the same emissions is stable at around 175 km of altitude, but the emissions from O are also occurring over a broader range of altitude than is the case for the 4278 Å and 6730 Å emissions. It is also possible to see





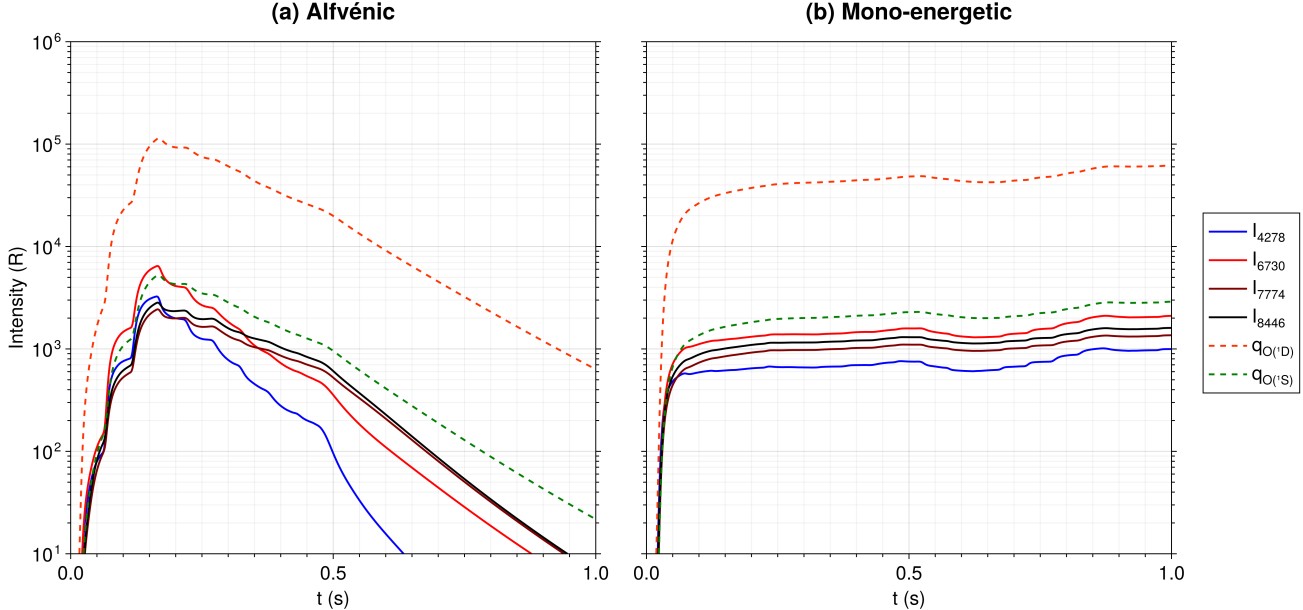

**Figure 5.** Column-integrated emission rates for 4278 Å, 6730 Å, 7774 Å and 8446 Å, obtained by integrating in height the volume emission rates seen in Fig. 4. The finite speed of light is taken into account. The intensities are plotted in units of Rayleigh. The intensities of $O(^1D)$ and $O(^1S)$ (dashed lines) are not column-integrated emission rates but column-integrated excitation rates. (a) Intensities calculated from the `aw1` event. (b) Intensities calculated from the `inv` event.

from Fig. 4 that for both events, the emissions at 6730 Å are stronger than at the other wavelengths. This difference in strength between the emissions seems to be more pronounced for the `aw1` event than for the `inv` event.

Integrating these optical emissions rates in height and taking into account delays due to the finite speed of light, one can calculate the expected optical intensity that would be seen on the ground. This is done and shown in Fig. 5. Note that the $O(^1D)$ and $O(^1S)$ intensities (dashed lines in Fig. 5) are not what would be seen by cameras on the ground. Since these emissions

are not prompt emissions, what is shown here are their excitation intensities. To calculate their volume emission rates and the resulting intensities seen from the ground, one would need to take the excitation rates calculated here and include them into a full ion chemistry model, with would take into account radiative decay, quenching, diffusion and drift effects. This is out of the scope of this article.

The differences between the precipitations of the `aw1` and `inv` events are clearly visible here. The column-integrated

optical intensities produced by the Alfvénic precipitation of event `aw1` present a clear peak at the beginning followed by a steady decrease in time, while the optical intensities produced by the steady precipitation of the `inv` event are very stable. Furthermore, there are differences in which wavelengths are dominant. For the mono-energetic case, the intensity order is stable in time from 0.075 s after the start of the precipitation. The order from the brightest to the faintest is $I_{6730} > I_{8446} >$



$I_{7774} > I_{4278}$. For the Alfvénic precipitation, $I_{6730}$ dominates at the peak, with $I_{4278}$ having the second highest intensity. After
the peak, both decrease faster than $I_{8446}$ and $I_{7774}$, which start to dominate after 0.35 s.

## 3.4 Comparison between observed and modeled upward flux

In this section, we perform a comparison between the observed and modeled upward electron flux at the altitude of the rocket. This amounts to an experimental first-order verification of AURORA – if the time and energy variations of the upward electron flux modeled with AURORA do not agree with the in-situ observations, the model would be disproved. Ideally, comparisons between optical emission rates produced by the code, such as the ones shown in subsection 3.3, with ground-based optical observations would also have been performed. However, due to the rocket trajectory over the north Atlantic ocean and partially cloudy conditions, it is challenging to perform such a comparison.

AURORA takes as input only the measured downward electron flux (pitch-angles of 0–90°) at the top of the simulation. This makes it possible to test AURORA by comparing the upward electron flux (pitch-angles of 90–180°) calculated by AURORA at the top of the simulation with the upward flux measured by the rocket. The dispersed structure seen in the `aw2` event is used for the comparison because it is relatively isolated. Indeed, it is preceded in the data by a few seconds of very little precipitation as can be seen in the row (e) of Fig. 1. This is good for making comparisons, since there is little contamination in the upward flux measured by the rocket from precipitation before the start of the `aw2` event.

The upward flux of the `aw2` event measured by the rocket are shown in Fig. 6a. Different panels correspond to different pitch-angle streams. Previously in this article, we described the upward electron flux has having pitch-angles from 90° to 180°, where 180° corresponds to field-aligned up. For easier identification, the pitch-angles in Fig. 6 are given from 0° to 90° with a suffix "UP", where 0° corresponds to field aligned up. Note that we do not use the electron flux with pitch-angles of 80–90° UP for the comparison. This is because one side of the instrument did not provide any measurement, and the other side did not separate between electrons with pitch-angles of 80–90° UP and 80–90° DOWN.

The upward flux modeled at the rocket altitude when using the downward flux of event `aw2` as input are shown in Fig. 6b. For better vizualisation, Fig. 6c, shows the contours of the colormaps from Fig. 6a and Fig. 6b superimposed. Comparing the measured and modeled upward flux, a few minor discrepancies can be detected:

1. The first difference is that the modeled flux are higher than the measurements at low energy (under 20 eV). This could be associated with instrumental effects, such as from the spacecraft potential or the lower sensitivity of top-hat ESA instruments at low energies. In particular, the rocket measures low flux at these energies during the whole flight. Furthermore, discrepancies could potentially also arise from physical processes that trap low energy electrons in the ionosphere not accounted for in AURORA.

2. The second difference is in the time of arrival of flux with a high pitch-angle (over 60°). These simulated flux arrive a tenth of second later than the measurements. Potential explanations could be the absence of magnetic mirroring physics in the code and/or ambiguities due to the rocket motion.





**Figure 6.** Electron energy upward flux measured by the rocket and calculated by the transport code. Different columns correspond to different pitch-angles of the electrons. (a) Electron energy upward flux as measured by the rocket during the `aw2` event. (b) Electron energy upward flux modeled by AURORA using the downward flux from the 536 s event as input. (c) Superposition of the contour lines from (a) and (b). Dashed lines are contours from (a), solid lines are contours from (b).



However, the agreement between the flux in the different panels in Figure 6 shows that AURORA is able to accurately reproduce the time–energy variations and the intensity of the upward electron flux measured in-situ by the rocket, especially for pitch-angles between 0° and 60° and energies above 20 eV. This suggests that AURORA adequately models the propagation, scattering and degradation in energy of the electrons in the ionosphere.

## 4    Discussion

In this study, we presented the time-dependent modeling of electron flux in the ionosphere. The results were obtained using AURORA, a recently developed multi-stream time-dependent electron transport code (Gustavsson, 2022), that was modernized and improved for the purpose of this study (Gavazzi and Gustavsson, 2025). The model was fed with precipitating high-resolution in-situ data from one of the VISIONS-2 sounding rocket that measured time–energy dispersed structures in the electron flux varying on sub-second time-scales.

Simultaneously, the rocket observed intense fluctuations in the perpendicular electric field and magnetic field components, a high downward Poynting flux, as well as the presence of vortex structures in the perpendicular electric field (Fig. 1). These observations support the hypothesis of the Alfvénic origin of the precipitation (e.g. Stasiewicz et al., 2000; Kletzing and Hu, 2001; Andersson et al., 2002; Miles et al., 2018; Pakhotin et al., 2020), a type of precipitation frequently observed in the dayside cusp (Chaston et al., 2002).

With the AURORA code, we modeled some effects of this kind of highly dynamic precipitation on the ionosphere for selected events. We showed that AURORA was generally able to reproduce the upward electron flux measured by the rocket, strongly supporting the ability of AURORA to model the electron flux in the ionosphere for the kind of dynamic precipitation observed during in the VISIONS-2 campaign. For further validation, future studies comparing modeled optical emission rates with high-resolution optical observations could be performed.

The ionospheric responses to Alfvénic and steady mono-energetic precipitation were modeled. First, we used the time–energy dispersed structure `aw1` (Fig. 1) as input. From the simulated electron flux, we calculated the evolution in height and time of the ionization rates (Fig. 3) and of the optical emission rates (Fig. 4). We observed that the peak of the ionization rate and of the various optical emission rates shifts in height through time. This is a known feature of dynamic aurora, particularly of a phenomena often referred to as "flaming aurora" (Omholt, 1971; Dahlgren et al., 2013). This type of aurora is believed to be the optical manifestation of Alfvén wave precipitation and associated time–energy dispersed structures in the electron flux (Semeter et al., 2008; Dahlgren et al., 2013), an hypothesis that our observations and modeling support. By running the simulations for a few extra tenths of seconds after the primary precipitation at the top of the ionosphere stopped, we were also able to show that the enhanced ionization and optical emissions continue for a few tenth of seconds after the precipitation stops. This is most likely due to the propagation of secondary electrons that continue to propagate and collide with the neutral atmosphere, a phenomena we are able to model with the time-dependent code.

For comparison we also modeled the mono-energetic precipitation event `inv` observed a few seconds after the dispersed structure `aw1` (see Fig. 1) and with a similar precipitating energy flux. Variations in the maximum intensity (Fig. 4) are percep-



tible, but the height of the peak intensity is stable over time. This is different from the dispersed structure aw1 described above
which deposits its energy over a wider range of altitudes. Furthermore, the column integrated optical emission were observed
to be relatively stable in time for mono-energetic precipitation, with the wavelengths 6730 Å and 8446 Å dominating, while for
Alfvénic precipitation, a peak at 6730 Å and 4278 Å was first observed followed by a rapid decay of these wavelengths. The
modeled intensity variations could be used in future optical studies to help characterize the precipitation spectra. An in-depth
quantitative analysis of the differences between the different precipitation types is left for future work. Such a study could be
combined with a comparison with high-resolution optical measurements.

## 5  Conclusions

Electron transport codes have been used in the past to advance our understanding of auroral precipitation (recently for example
Lynch et al., 2007; Grubbs II et al., 2018; Gabrielse et al., 2021; Yu et al., 2022). Here, we introduced a new version of
AURORA, a recently developed time-dependent electron transport code (Gustavsson, 2022; Gavazzi and Gustavsson, 2025).
The work presented here shows the first results and verification of using the code with high resolution in-situ data to model
auroral precipitation on sub-second time-scales.

Alfvén waves were identified in the in-situ electric and magnetic data from one of the VISIONS-2 sounding rocket. We
used the associated precipitation, seen as time–energy dispersed structures in the electron flux measurements, as input to the
transport code. We also used mono-energetic precipitation as input and presented differences in the ionization and emission
rates associated with both types of precipitation. For the Alfvén wave case, the maximum intensity of the ionization and
emission rates changes rapidly over a wide range of altitudes, while for the mono-energetic case, the maximum intensity is
stable in height. We also found that the wavelengths that dominate the emissions are different for the two cases and are seen to
vary in time for the Alfvénic precipitation.

As suggested by Sandahl et al. (2011), time-dependent modeling of precipitation opens for possibilities to advance our
understanding of different aspects related to small-scale dynamic aurora. In particular, the next step involves performing a
statistical study by characterizing various dynamic precipitation events, such as a train of waves or individual waves with
different time–energy slopes, and investigate and quantify their effects on the ionosphere. Another possibility is to build on
the technique presented here to study variations of other ionospheric parameters, such as the conductivity or the heating,
in response to dynamic precipitation. Additionally, combining time-dependently modeled optical emission rates with optical
measurements opens possibilities to further study dynamic auroras, such as flickering or flaming aurora.

*Code and data availability.*  Figures were made using *Makie.jl* (Danisch and Krumbiegel, 2021). Scripts and data to reproduce the simulation
results and figures of the paper are available at https://doi.org/10.5281/zenodo.15341639. The MSIS2.1 (Emmert et al., 2021) model data
used in AURORA was obtained using the *pymsis* python wrapper (Lucas, 2022), available at https://github.com/SWxTREC/pymsis. The
IRI2016 (Bilitza et al., 2017) model data used in AURORA was obtained using the *iri2016* python wrapper (Ilma, 2017), available at
https://github.com/space-physics/iri2016.



*Video supplement.* An animation of the ionospheric flux is provided as supplementary material.

*Author contributions.* EG and AS conceptualized the study. EG performed the simulations, validated and analyzed the results, and created all the figures. EG and BG developed the software (AURORA). AS provided supervision. DR is the PI of the rocket. JC and RP provided the electron flux data and the electric/magnetic field data, respectively, and assisted with interpretation. EG prepared the original draft with contributions from AS. All co-authors reviewed and edited the final manuscript. All co-authors contributed to scientific discussions around the project.

*Competing interests.* The authors declare that they have no conflict of interest.

*Acknowledgements.* A. Spicher acknowledges funding from the Research Council of Norway (RCN) grant 326039, and the UiT The Arctic University of Norway contribution to EISCAT_3D (RCN funded grant 245683). J. Clemmons gratefully acknowledges support of the VISIONS-2 experiment by NASA through Grant NNX16AF02G. D. Rowland acknowledges support from the NASA ROSES Heliophysics Technology and Instrument Development for Science program and the NASA Sounding Rocket Program. R. Pfaff and D. Rowland gratefully acknowledge support from NASA's Science Mission Directorate which enabled these experiments to be carried out.



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
