# Peer review of "Time-dependent modeling of Alfvénic precipitation observed in the ionosphere"

_EGUsphere, 2025_

## Author Comment (AC2)

We thank the reviewers for their positive and constructive feedbacks. Here we address their comments. The comments from the reviewers are in bold font, and our answers in blue font.

**Main points:**

In the simulations, the number is grid elements in energies and altitudes is never mentioned. It is the same for the limits of these grids. For example, line 129, an energy of 5 keV is mentioned. Is it the higher energy considered? The kind of grids must also be clarified. Are they linear or exponential or built in another way?

We thank the reviewer for pointing this out. We agree that these details are important and should be presented. As such, we will do the following modifications to the text in the revised manuscript (section "2. Instrumentation and simulation") to give more information about the grids and their limits. This new text also addresses other comments from the reviewer.

**The paragraph**

"AURORA is run with the observed electron flux,  $Ie(E, t, \theta)$  as input at the highest altitude. 18 pitch-angle streams uniformly spaced from 0° to 180°, each with a width of 10°, were used to best model the pitch-angle distribution. The energy and pitch-angle grid of the observations were up-sampled to match the default grid of AURORA. The Ie observations were made with a time-resolution of 50 ms and are taken as constant over each interval. To properly resolve time-of-flight effects for the fastest electrons, at 5 keV, AURORA integrates the transport equations with time-steps smaller than 500  $\mu$ s."

**will be rewritten as**

"To solve equation (1) numerically, AURORA introduces grids in altitude, energy, pitch-angle and time. The description of the grids that follows applies to all the simulations presented in this paper. The altitude domain spans from 100 km to the rocket altitude at the time of measurements, which is around 600km for the cases shown here. The altitude grid consists of 410 points with finer spacing at lower altitude to resolve steep gradients due to shorter collision mean free paths. In numbers, the step size varies from 150 m at 100km to 10 km at the top. The energy domain spans from 2 eV to 5 keV, which is a few keV above the maximum energy of the precipitation events that are considered. The energy grid consists of 508 elements and is piecewise non-uniform. Above 500 eV, it uses a constant step size of 11.65 eV, which is smaller than the lowest ionization threshold for the three major species considered: N2, O2 and O. Below 500 eV, the step size is reduced smoothly down to 0.15 eV at 2 eV in order to resolve the variation of electron fluxes at energies around the thresholds of inelastic collisions. The pitch-angle grid uses 18 streams uniformly spaced from 0° to 180°, each with a width of 10°. Time integration is done with a fixed time step of 333 μs, which, based on testing, appears to be a good trade off between runtime and accuracy for the fastest electrons considered (5 keV).

AURORA is run with the observed electron flux,  $Ie(E, t, \theta)$  as input at the highest altitude. The energy and pitch-angle grid of the observations are up-sampled to match the grids of the simulations. The simulations are initialized without any electron flux in the ionosphere and the first observations of precipitation are used as initial flux at the highest altitude. As the time resolution of the observations (50 ms) is coarser than the time-step used in the simulations (333  $\mu$ s), the input flux are treated as piecewise constant in time and are updated every 50 ms to the next measurement. We opted against temporal smoothing to avoid introducing arbitrary interpolation"

For the altitude, it also exists an ambiguity: Fig. 2 shows simulations up to 600 km, but it is mentioned line 206 that the top of the modeled ionosphere is above 400 km. Please clarify.

The text we will add to the revised manuscript in section "2. Instrumentation and simulation" (see answer to previous comment) helps clarify this apparent ambiguity: the top altitude for the simulations is always chosen to be equal to the rocket altitude at the time of the measurements (around 600km for the three events).

To avoid confusion, we will also modify the following text line 206:

"Indeed, the high energy electrons (> 100 eV) are seen to reach altitudes below 300 km, while the lower energy part of the precipitation (< 100 eV) is just arriving at the top of the modeled ionosphere, above 400 km."

into

"Indeed, the high energy electrons (> 100 eV) are seen to reach altitudes below 300 km, while the lower energy part of the precipitation (< 100 eV) is just arriving above about 400 km."

In the simulations also and considering that the time step is much shorter than the measurement time resolution, the initial conditions are not clear. Is it the one shown in the top panel of figure 2 or the precipitation during the previous measurement time bin?

The procedure is described in more details in the text that will be added in section 2 (see answer to first comment).

It could also be interesting to described how the precipitation input is managed regarding the data. Is it smoothed in time or do the authors change them abruptly?

The procedure is described in more details in the text that will be added in section 2 (see answer to first comment).

The choice of the time stamp in the simulations presented in figure 2 (0.041,....) are not justified. Is it arbitrary or does it have a specific justification? Please clarify.

The time stamp for the time slices of figure 2 were chosen arbitrarily to exemplify how the flux are evolving in the ionosphere. We will clarify this in the revised text.

Line 188: "In Fig. 2 are shown snapshots at 0.041 s, 0.129 s and 0.323 s after the start of the event. These time stamps were chosen arbitrarily to show the evolution of the flux in the ionosphere. Here we have [...]"

It is similar for the time step. On line 129, a step of less than "500  $\mu$ s" is mentioned. Did the authors use the same time step in the shown simulation or a smaller one?

The time step used in the shown simulations is the same as the one indicated in section "2. Instrumentation and simulation". This is clarified in the text that will be added in section 2 (see answer to first comment).

In figure 3, the simulated precipitations are stopped at 0.45 s in the Alfevenic case in order to visualize the decrease of the ionization rates. The authors must justify why they do this in the Alfvenic case and not in the other case (quasi mono energetic).

In this manuscript, we focus on modeling dynamic auroral precipitation. As this type of precipitation typically comes in the ionosphere as a "burst", we considered it interesting to see how the ionization rates would decrease after such an event. Thus, we let the simulation continue running with no precipitation.

We also compared the results with a quasi mono-energetic event to highlight differences in ionization and optical emission rates profiles, such as the ones that would be observed by cameras on the ground for these two types of precipitation. We did not let the simulation run without precipitation after the quasi mono-energetic event as we considered it unlikely that this kind of

precipitation would stop abruptly. This is supported by the full-flight data from the VISIONS-2 rocket, where we do not observe a sudden-stop behaviour for this type of precipitation.

We will add this clarification in the text at line 221, at the end of the paragraph:

"[...] consists of a subset of a mono-energetic structure stable for several seconds. Given the bursty nature of Alfvénic precipitation, we extended the aw1 simulation by an additional 0.55 s with no input in order to visualize the ionosphere's relaxation after the precipitation ceases. This was not done with the inv event, since a sudden cutoff of quasi mono-energetic precipitation is not observed in the rocket data and appears rather unlikely for quasi-static potential structures."

**Then, at line 226, the text**

"[...] at around 160 km. The simulation for the aw1 event was run for an extra 0.55 s without any precipitation to show what is happening after the precipitation stops. We observe that [...]" will be replaced by

"[...] at around 160 km. After precipitation stops in the aw1 event simulation, we observe that [...]"

We will also add vertical lines in the panels in figures 4a and 5a to indicate the end of the precipitation. The captions will be updated adequately.

**Last major point: It is not clear why, in Figure 6, the simulations after 0.45 s are different from the observations for all pitch angles. This discrepancy is not mentioned in the text. Please clarify.**

We thank the reviewer for this comment. We forgot indeed to explain this discrepancy. A plausible explanation could be that the rocket is moving in space out of the magnetic flux tube where the precipitation was taking place. As the rocket moves out of the flux tube, it stops measuring the albedo flux produced by the precipitation. This hypothesis is consistent with the observation that the sudden drop in up-flux intensity in the rocket data occurs simultaneously in all pitch-angles. We will add the following text in the revised manuscript to underline and explain this discrepancy. "3. The third difference is that the observed flux intensity drops for all pitch-angles after 0.5 s, while they do not for the simulated flux. This could be an effect of the spatial motion of the rocket moving out of the magnetic flux tube along which the electrons are precipitating."

It is worth mentioning here that this figure has received other changes. See the end of this document for additional information.

**Minor other points:**

The term "mono-energetic" is confusing since the precipitations are not fully mono-energetic. Perhaps quasi mono-energetic should be closer to the reality.

We agree with the reviewer. The term "mono-energetic" will be changed to the suggested "quasi mono-energetic" in the revised manuscript.

In lines 166 and 170, it is mentioned twice, "which is believed to be the sign of DAWs followed by references. It could be useful, behind the references, to explain in a few sentences what the phenomenology behind these possibilities.

We will change the following text in the revised manuscript:

**Line 165**

"These intense fluctuations in the perpendicular field components can be a sign of the presence of dispersive Alfvén waves (DAW) (Stasiewicz et al., 2000; Miles et al., 2018; Pakhotin et al., 2020)" into

"These intense fluctuations in the perpendicular field components over a wide range of frequencies are the sign of electromagnetic wave activity. As such, they are often interpreted as the

manifestation of dispersive Alfvén waves (DAW) (Stasiewicz et al., 2000; Miles et al., 2018; Pakhotin et al., 2020)"

Line 170

"These vortices are believed to be caused by DAWs (Stasiewicz et al., 2000)." into

"These vortices indicate electromagnetic wave activity with circular/elliptical polarization and have been observed in regions with small-scale (<= 1km) DAWs (Stasiewicz et al., 2000)."

Line 140: It should be mentioned here that "prompt" emissions make the difference between allowed and forbidden transitions in the dipolar electric approximation and that it excludes the green and red lines, which are long-lived transitions and thus forbidden. We will change the text from

"Similarly, one can calculate the excitation rates of different excited states of the neutrals, given the corresponding excitation cross-sections. If the relaxation from these states lead to prompt optical emissions, then the excitation rates are equivalent to optical emission rates. If the excited states have non-negligible lifetimes, the full ion chemistry with effects from radiative decay, quenching, diffusion, drift, etc. should also be taken into account to obtain the optical emission rates. This is out of the scope of this article, and here we limit ourselves to prompt emissions."

**into**

"Similarly, one can calculate the excitation rates of different excited states of the neutrals, given the corresponding excitation cross-sections. If the relaxation from an excited state leads to a photon emission that is allowed in the electric dipole approximation (Rees, 1989, p 148), also called a prompt optical emission, then the excitation rate of this state is equivalent to the optical emission rate of the associated spectral line. If the relaxation is a forbidden transition, the excited state has a non-negligible lifetime of a few seconds to minutes, and the full ion chemistry with effects from radiative decay, quenching, diffusion, drift, etc. should be taken into account to obtain the corresponding optical emission rate. The green (O1S) and red (O1D) lines in the aurora are such forbidden transitions (Rees, 1989, p 152-154). In this article, we limit ourselves to calculating the emission rates of prompt emissions."

**Line 189: Why the 10-20° and 20-30° pitch angles are merged. This must be justified.**

This is to reduce the number of panels and make the figure more readable. This precision will be added in the revised text.

Line 189:

"[...] start of the event. To reduce the number of panels in the figure, we have merged some pitchangle streams [...]

**Lines 222 to 229 and relation with figure 3: It should be mentioned in the text that the red line is the altitude of the emission peak.**

This will be added to the revised text.

"This is done for both the aw1 and inv events, and the resulting ionization rates are shown in Fig. 3. The height of maximum ionization rate, or peak intensity, is indicated by a red dashed line in both panels. The peak intensity [...]"

In the legend of figure 4, please give the full spectroscopic notation of the state and not only B3. Mention also that it is the v'-v 5-1 transition of this band and not the full band. The same goes for the 427 band, which is the 0-1.

We changed to "4278 Å emitted by the N2+ 1N (0, 1) band emissions," "6730 Å by the N2 1PG (5, 2) and (4, 1) band emissions,"

The paragraph between lines 242 and 248 is confusing; it is mentioned above that the forbidden transitions are not taken into account. The authors could remove it or better justify why they mention these two emissions lines here.

We agree with the reviewer that this was confusing. We will remove the forbidden transitions from figure 5, its caption, and the associated text in the revised manuscript.

**Figure 6: A small arrow appears in the left top panel (case a, angle 0-10°). Does it have a signification?**

We believe that the "arrow" the reviewer is referring to is an unfortunate contour line and its associated label whose combined shape resembles an arrow. To avoid confusion, we removed the labels of the contour lines in that first row.

Line 288-290: I think it is necessary to be more prudent when saying that simulations reproduce the observation. I suggest "reproduce in many cases the observations but not all of them."

We agree with the reviewer and will modify the text in the revised manuscript as follow: "[...] is able to generally reproduce the time—energy variations and intensity of [...]."

**Line 330 : Please be more precise when writing "high resolution." Does it concern space or time. resolution or both?**

We meant that the rocket instrument has a high temporal sampling rate. Consequently, the spatial resolution is also high. We will modify the text to "using the code with in-situ data gathered with a high sampling rate to model [...]"

**Additional information and change**

During the time of review, we identified and revised a pitch-angle indexing offset in the rocket electron flux data. We reran the simulations with the updated data, and the revised figures are included here. No other changes were made to the analysis or simulations configurations. Values for the ionization and emission rates have changed slightly, but the qualitative analysis and the conclusions of the paper are unchanged. The agreements of up-flux in figure 6 have actually improved.

The Zenodo repository (<a href="https://doi.org/10.5281/zenodo.15341639">https://doi.org/10.5281/zenodo.15341639</a>) associated with this paper will be updated with the new functions that were used to reprocess the rocket data.

To reflect the changes done to the raw rocket data, we will add text in the revised manuscript. At line 100, end of the paragraph:

"[...] lower measurement intensity. Prior to averaging, we adjust a minor pitch-angle indexing offset in the instrument data and rescale with the associated solid-angle weights. After this correction, the two hemispherical measurements 0--180° and 180--360° are nearly symmetric in shape and intensity, and transient enhancements of precipitation are observed to arrive field-aligned first."

**Updated Figure 2:**

**Updated Figure 3:**

**Updated Figure 4:**

**Updated Figure 5**

**Updated Figure 6**